

# Phase separation in binary Bose mixtures at finite temperature

Gabriele Spada[1][⋆], Luca Parisi[2], Gerard Pascual[3], Nicholas G. Parker[2],
Thomas P. Billam[2], Sebastiano Pilati[4,5], Jordi Boronat[3] and Stefano Giorgini[1]

**1** Pitaevskii Center on Bose-Einstein Condensation, CNR-INO and
Dipartimento di Fisica, Università di Trento, 38123 Povo, Trento, Italy
**2** Joint Quantum Centre (JQC) Durham-Newcastle, School of Mathematics, Statistics and
Physics, Newcastle University, Newcastle upon Tyne, NE1 7RU, United Kingdom
**3** Departament de Física i Enginyeria Nuclear, Universitat Politècnica
de Catalunya, Campus Nord B4-B5, E-08034, Barcelona, Spain
**4** School of Science and Technology, Physics Division,
Università di Camerino, 62032 Camerino, Italy
**5** INFN, Sezione di Perugia, I-06123 Perugia, Italy

⋆ gabriele.spada@unitn.it

## Abstract

We investigate the magnetic behavior of finite-temperature repulsive two-component Bose mixtures by means of exact path-integral Monte-Carlo simulations. Novel algorithms are implemented for the free energy and the chemical potential of the two components. Results on the magnetic susceptibility suggest that the conditions for phase separation are not modified from the zero temperature case. This contradicts previous predictions based on approximate theories. We also determine the temperature dependence of the chemical potential and the contact parameters for experimentally relevant balanced mixtures.



# 1 Introduction

The realization of mixtures of ultracold gases in the quantum-degenerate regime has opened new interesting directions to study the simultaneous presence of superfluidity in multicomponent systems, which could not be addressed with traditional quantum fluids such as liquid $^4$He or standard superconductors. The first examples of superfluid mixtures have been produced both with atoms obeying the same [1, 2] or different statistics [3]. In particular, two-component mixtures of bosonic species below the Bose-Einstein transition temperature provide one with the simplest set up to investigate the interplay between quantum magnetism and superfluid properties. This includes novel phenomena such as combined mass and spin superfluidity [4], non dissipative spin drag [5], and Bose-enhanced magnetic effects [6]. In the case of repulsive mixtures, the zero temperature scenario is well described by mean-field theory [7]: the ground state is paramagnetic if the interspecies coupling constant is below a threshold set by the strength of interactions within each component, and is instead fully ferromagnetic, i.e. full phase separation between the two components occurs, if the coupling exceeds this critical value. This scenario has been also confirmed in a series of experiments [8–11] and quantum Monte Carlo simulations for trapped mixtures, both at zero [12] and finite temperature [13]. At finite temperatures, perturbative approaches, such as Hartree-Fock (HF) and Popov theories, predict an intriguing scenario holding for mixtures below the Bose-Einstein condensation (BEC) temperature: The paramagnetic state at low temperature can turn ferromagnetic at higher temperature if the interspecies coupling is close enough to the $T = 0$ threshold [14–16]. According to these theoretical schemes, the mechanism responsible for the magnetic transition are beyond mean-field effects induced by temperature, which destabilize the paramagnetic phase. Similar effects of pure quantum nature have instead a stabilizing role in attractive mixtures and lead to the formation of self-bound droplets [17–19]. An important question, which needs to be answered, is whether the predictions of perturbative approaches are accurate enough to include the relevant role played by fluctuations around the transition temperature.

In this work we use exact path-integral Monte Carlo (PIMC) simulations to investigate the magnetic and thermodynamic properties of a repulsive two-component Bose mixture. In particular, novel algorithms are implemented to obtain precise unbiased predictions for the chemical potentials of the two separate components and for the total free energy. This provides us with crucial information on the chemical equilibrium at finite polarization and on the occurrence of stable free energy minima. We find that the magnetic susceptibility at finite temperature is well described by the simple zero temperature mean-field prediction and also that there is no evidence of a temperature-induced ferromagnetic transition. Consequently, the conditions for phase separation remain unchanged from the $T = 0$ case. Furthermore, for the choice of interspecies coupling corresponding to a balanced mixture of sodium atoms, we calculate chemical potential and contact parameters as a function of temperature, pointing out their deviations in the critical region from the predictions of perturbative methods. In partic-

ular, the interspecies contact parameter features a suppression at intermediate temperatures caused by statistical effects which indicates an enhanced repulsive correlation between the two components.

## 2 Methods

We consider the following Hamiltonian describing a system of $N = N_1 + N_2$ Bose particles belonging to two distinguishable components with equal mass $m$

$$H = -\frac{\hbar^2}{2m}\sum_{i=1}^{N_1}\nabla_i^2 - \frac{\hbar^2}{2m}\sum_{i'=1}^{N_2}\nabla_{i'}^2 + \sum_{i<j}^{N_1}v(|\mathbf{r}_i - \mathbf{r}_j|) + \sum_{i'<j'}^{N_2}v(|\mathbf{r}_{i'} - \mathbf{r}_{j'}|) + \sum_{i,i'}^{N_1,N_2}v_{12}(|\mathbf{r}_i - \mathbf{r}_{i'}|). \quad (1)$$

The intraspecies potentials are assumed to be the same, denoted by $v(r)$, and $v_{12}(r)$ describes interspecies interactions. All potentials are repulsive and modeled by hard spheres, *i.e.* the potential is infinite inside the diameter of the sphere and zero outside. The two parameters $a$ and $a_{12}$ define, respectively, the range of the $v$ and $v_{12}$ potential and the corresponding value of the $s$-wave scattering length. In the dilute regime of interest, interaction effects only depend on the coupling strengths: $g = \frac{4\pi\hbar^2 a}{m}$ and $g_{12} = \frac{4\pi\hbar^2 a_{12}}{m}$. To discuss magnetic properties we introduce the component densities $n_1 + n_2 = n$ and the polarization parameter $p = (n_1 - n_2)/n$. For such a symmetric mixture, mean-field theory at zero temperature predicts miscibility ($p = 0$) if $g_{12} < g$ and a fully separated state ($p = 1$) if $g_{12} > g$ [7]. Furthermore, the same theory yields the expression $\chi = \frac{2}{g - g_{12}}$ for the magnetic susceptibility in the paramagnetic phase.

In a PIMC simulation, we use periodic boundary conditions in a box of volume $V$ at fixed density $n = N/V$. We work with the well established worm algorithm in continuous space to efficiently sample bosonic permutations [20]. Recently the method has been further developed to be fully consistent with periodic boundary conditions and was applied to the study of the single-component gas [21]. The algorithm is described in details in Ref. [22]. In the present study, we implemented also the calculation of the total free energy, of the free energy differences for different polarizations, and of the chemical potentials for both components in the canonical ensemble, generalizing to mixtures the technique first proposed in Ref. [23]. The details of the Monte Carlo moves added to the PIMC algorithm can be found in the Appendices A and B. In addition, we use also HF and Popov theories to compare with PIMC results. Details on the derivation of the free energy and related quantities within the HF and Popov scheme are given in Appendix C.

## 3 Magnetic behavior of binary mixtures

We first focus on the magnetic properties of the mixture, analyzing how the chemical potential and the total free energy depend on the polarization. We choose the value $na^3 = 10^{-4}$ for the gas parameter which on one side emphasizes the interesting effects due to interactions and on the other side ensures that the results are universal in terms of solely the gas parameter. However it is worth pointing out that stronger interactions could be realized in resonantly interacting gases [24].

In Fig. 1, the chemical potentials of the two components are plotted against polarization at fixed temperature $T = 0.794 T_c^0$, where $k_B T_c^0 = \frac{2\pi\hbar^2}{m}(n/2\zeta(3/2))^{2/3}$ is our reference energy scale, corresponding to the BEC transition temperature of a balanced ($p = 0$) non-interacting mixture. The majority component 1 is Bose condensed for all values of $p$ shown in the figure, while, at this temperature, the minority component 2 turns normal at the critical polarization

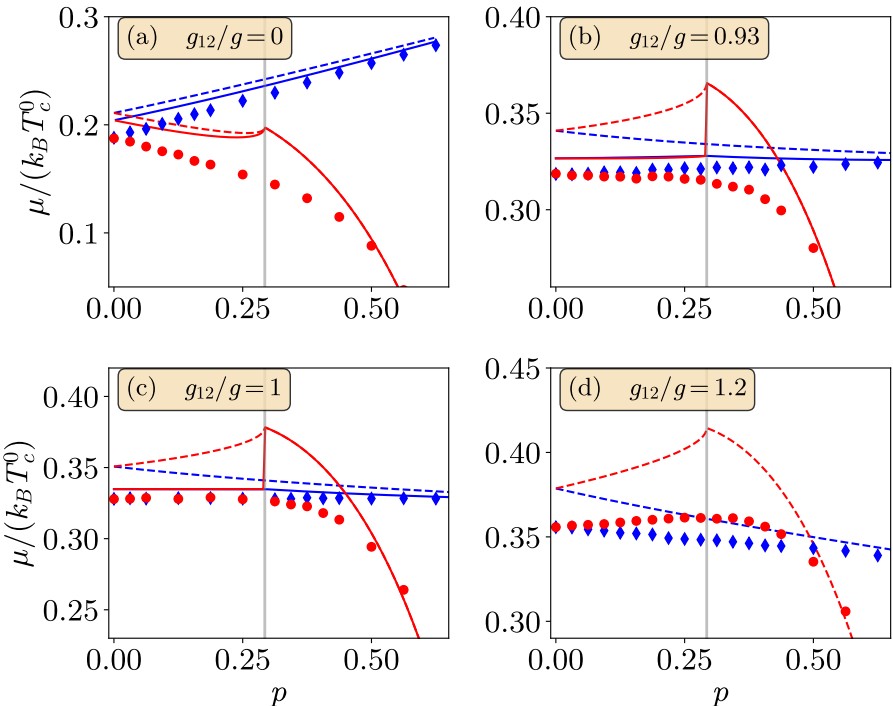

Figure 1: Chemical potentials $\mu_1$ (blue) and $\mu_2$ (red) as a function of the polarization $p = (N_1 - N_2)/N$ for a system with a total of $N = N_1 + N_2 = 128$ particles at temperature $T = 0.794 T_c^0$ and with gas parameter $na^3 = 10^{-4}$, for four values of the couplings ratio $g_{12}/g$. The dashed lines are the HF predictions, the solid lines are the Popov predictions. For the minority (red) component, the two coincide for $p > p_c$. In panel (d) only the HF lines are shown. The vertical lines indicate the critical polarization $p_c$.

$p_c \simeq 1 - (T/T_c^0)^{3/2} \simeq 0.292$ corresponding to the maximum of the HF and Popov results for $\mu_2$. In panels (b) and (c), referring to $g_{12} > 0$, we notice that HF and Popov theories predict a crossing of chemical potentials at finite polarization $p > p_c$. This crossing corresponds to a minimum in the free energy $F$ according to the thermodynamic relation $\mu_1 - \mu_2 = \left( \frac{\partial F/N}{\partial p} \right)_{n,T}$. The minimum signals the phase-separated state where the majority component is Bose condensed and in equilibrium with the minority one in the normal phase [15]. This behavior of the HF and Popov free energies is shown in Fig. 2 [see panels (b) and (c)]. Note that the chosen value of $g_{12}/g = 0.93$ corresponds to the $|F = 1, m_F = 1\rangle$ and $|F = 1, m_F = -1\rangle$ Bose-Bose mixture of $^{23}$Na atoms investigated experimentally in Refs. [4, 25]. According to HF and Popov theories, this mixture should provide an example of the striking phenomenon of a paramagnetic state at low temperature which turns ferromagnetic at higher temperatures, as predicted in Ref. [15]. However, the PIMC results for $\mu_1$ and $\mu_2$ at $g_{12}/g = 0.93$, do not confirm this scenario. The majority component chemical potential $\mu_1$ is in good agreement with the Popov result, but $\mu_2$ deviates significantly in the region $p > p_c$ and does not exhibit the peak predicted by HF and Popov theories. As a result, no crossing occurs for $p > p_c$ and no minimum appears in $F(p)$ other than at $p = 0$. Furthermore, from the thermodynamic relation $F(p) = F(0) + \frac{N}{2} \frac{np^2}{\chi}$ holding at small polarization, we find a good agreement using the zero temperature mean-field result $\chi = 2/(g - g_{12})$ of the magnetic susceptibility, as can be seen in panels (a) and (b) of Fig. 2, where the MF prediction, shifted to coincide with the PIMC data at $p = 0$, well reproduces the $p^2$ behavior of the PIMC data. Similar results are obtained for the fully symmetric case $g_{12} = g$, where the chemical potentials exactly coincide for $p < p_c$

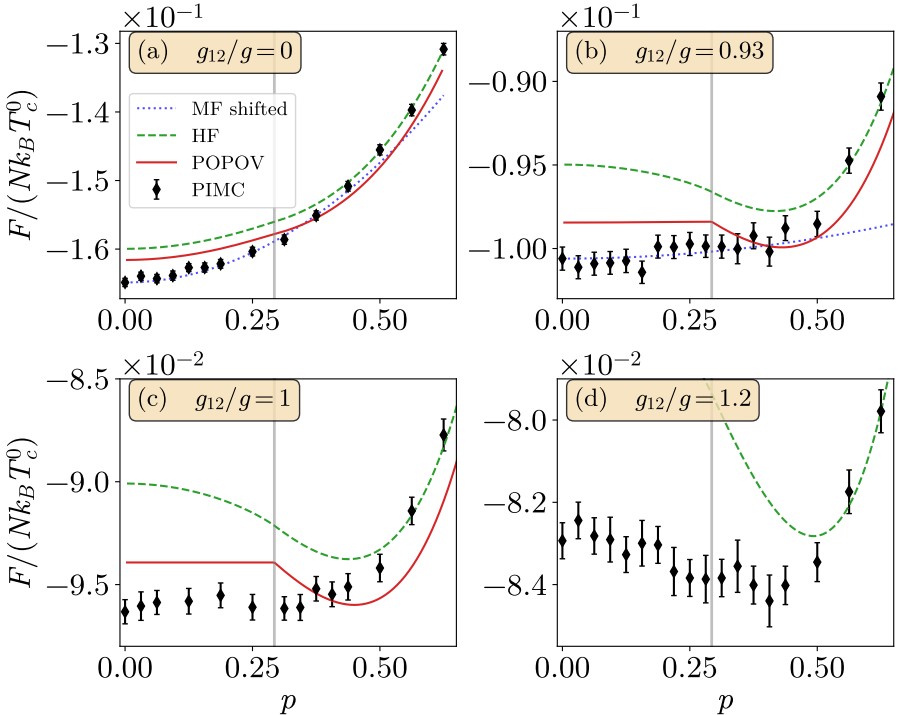

Figure 2: Free energy as a function of the polarization $p = (N_1 - N_2)/N$ for a system with a total of $N = N_1 + N_2 = 128$ particles at temperature $T = 0.794 T_c^0$ and with gas parameter $na^3 = 10^{-4}$, for four values of the couplings ratio $g_{12}/g$. The dotted blue lines in the panels (a) and (b) are the parabolas obtained from the mean-field prediction of the magnetic susceptibility $\chi = 2/(g - g_{12})$, the green dashed lines are the HF predictions, the red solid lines are the Popov predictions. In panel (d) only the HF line is shown. The vertical lines indicate the critical polarization $p_c$.

and separate without crossing for larger polarizations. As a result the free energy is flat as a function of polarization and the magnetic susceptibility diverges. Interestingly, this behavior, which is well understood at $T = 0$ where the ground state is degenerate with respect to polarization, remains valid at finite temperature as long as both condensates are present. The results shown in panels (a) and (d) of Figs. 1 and 2 are instead in better qualitative agreement with approximate perturbative approaches. The case $g_{12} = 0$ corresponds to no interaction between the two components: $\mu_2$ decreases with $p$, although without a small peak, and $F$ monotonically increases following the mean-field magnetic susceptibility. More interesting is the case $g_{12} = 1.2g$, where the mixture is phase separated already at $T = 0$. Notice that Popov theory can not be applied here if both condensates are present because spin excitations acquire an unphysical complex energy. The minority chemical potential $\mu_2$ displays a maximum, although not as large as predicted by HF theory, and a crossing point with $\mu_1$. As a consequence, the free energy indicates instability at $p = 0$ and shows a clear minimum at $p > p_c$, corresponding to the phase separated state with partial polarization.

We further analyze the magnetic behavior of the mixture in Fig. 3 where we show the free energy difference $\Delta F = F(p) - F(0)$ as a function of $p^2$ for the intermediate value $g_{12} = 0.5g$ of the interspecies coupling constant and at $T = 0.794 T_c^0$. This choice of parameters and, in particular, the choice of temperature emphasizes thermal effects in HF and Popov theories yielding important corrections to the $T = 0$ magnetic susceptibility. We also note that finite-size effects in PIMC simulations of the free energy are negligible if one increases further the total number of particles. We find that $F$ depends linearly on $p^2$ over a large range of values

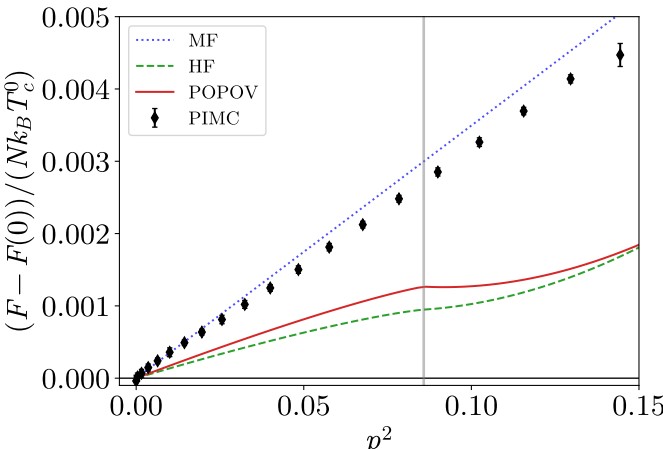

Figure 3: Free energy difference for a mixture with $na^3 = 10^{-4}$, and $g_{12}/g = 0.5$ and $T = 0.794 T_c^0$ as a function of the polarization squared. The PIMC results are compared with the $T = 0$ mean-field (MF - blue dotted line), HF (green dashed line) and Popov (red solid line) predictions. The vertical line indicates the critical polarization $p_c$.

extending also beyond the critical polarization $p_c$. Furthermore, the coefficient of the linear dependence, proportional to $\chi^{-1}$, is well reproduced by the mean-field result $\chi^{-1} = (g - g_{12})/2$ shown in the figure by the MF line. In contrast, HF and Popov results provide a poor account of the polarization dependence of the free energy. A possible explanation of this inadequacy involves the role of critical fluctuations which control the thermodynamics close to the transition point and, in general, can not be described using perturbative methods such as HF and Popov theories. The width of the critical region is predicted to shrink as $na^3 \to 0$ [26], but for experimentally relevant values of the gas parameter ($na^3 \simeq 10^{-4} - 10^{-6}$) it remains of the same order as the transition temperature itself.

From these results we conclude that, in contrast to HF and Popov predictions, the magnetic susceptibility depends very little on the temperature, and the conditions for phase separation seem to remain the same as at $T = 0$. In fact, if $g_{12} < g$, our results indicate that the only thermodynamically stable phase is the paramagnetic state at $p = 0$. A ferromagnetic state forms when $g_{12} > g$ and the effect of temperature is to reduce the equilibrium polarization from the $p = 1$ value achieved only at zero temperature. This is found at a high temperature not far from the BEC transition point and we expect the same to be true also for lower temperatures, where thermal effects not captured by the mean-field description should play a minor role. In this respect one should also notice that higher order interaction effects at $T = 0$ do not change the critical value $g_{12} = g$ for the onset of ferromagnetism (see Ref. [16]). As an additional remark, we point out that our results do not exclude a non trivial interplay between ferromagnetic and critical fluctuations in the close vicinity of the transition point. To carefully investigate these effects would require a much deeper analysis of the shift of the transition point in interacting mixtures beyond the scope of this work. Furthermore, we expect the simple $T = 0$ scenario to hold also at densities lower than $na^3 = 10^{-4}$. Numerical checks show that for vanishing gas parameter the free energy difference between the $p = 0$ state and the stable minimum at finite $p$ predicted by Popov theory is suppressed as $g^{3/2}$ and furthermore the minimum is shifted towards higher temperatures occurring closer to the transition point. As a consequence, we expect critical fluctuations to play a major role in the magnetic response of the mixture also in the regime of extremely low densities, thereby invalidating the predictions of Popov theory.

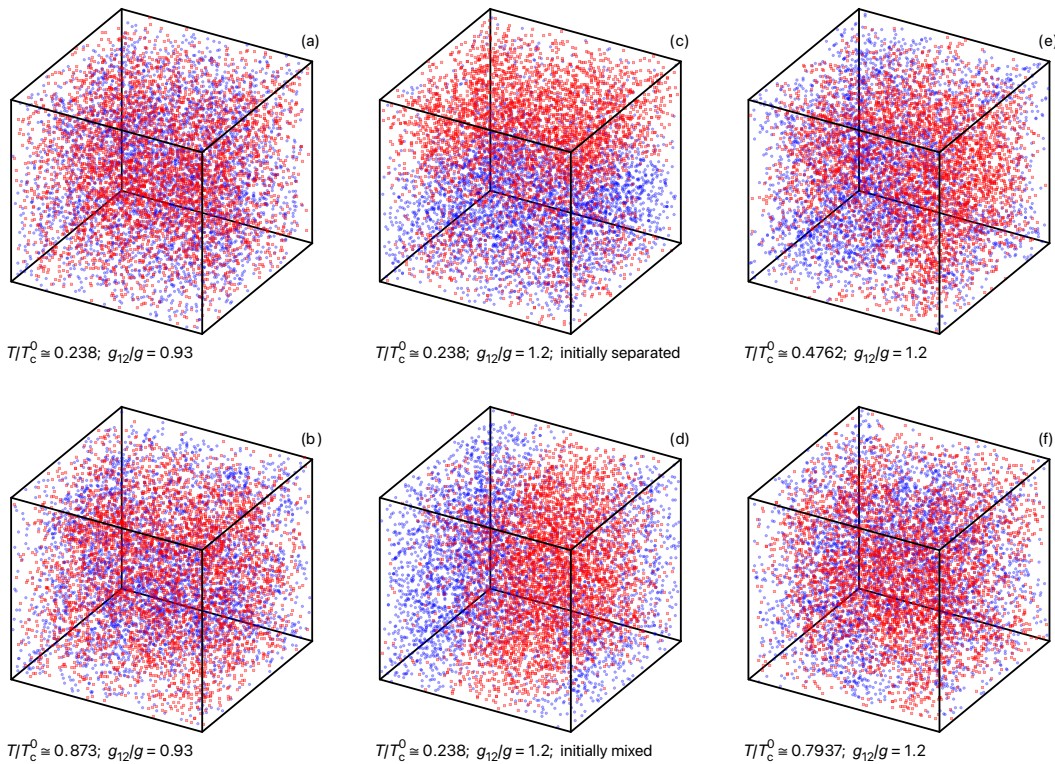

Figure 4: Snapshots of particle positions during PIMC simulations at equilibrium. Blue circles represent the $N_1 = 4000$ particles of the first component, the red squares represent the $N_2 = 4000$ particles of the second component. A single imaginary-time slice is considered. The gas parameter is $na^3 = 10^{-4}$. The five panels correspond to different temperatures $T$, interspecies couplings $g_{12}$, or different initial configurations. Panel (a): $T/T_c^0 \cong 0.238$, $g_{12}/g = 0.93$, components initially separated (along the vertical direction). Panel (b): $T/T_c^0 \cong 0.873$, $g_{12}/g = 0.93$, components initially separated. Panel (c): $T/T_c^0 \cong 0.238$, $g_{12}/g = 1.2$, components initially separated. Panel (d): $T/T_c^0 \cong 0.238$, $g_{12}/g = 1.2$, components initially mixed. Panel (e): $T/T_c^0 \cong 0.4762$, $g_{12}/g = 1.2$, components initially separated. Panel (f): $T/T_c^0 \cong 0.7937$, $g_{12}/g = 1.2$, components initially mixed.

## 3.1 Particle-position snapshots

Visualizing instantaneous particle positions during PIMC simulations allows us to shed some light on the ferromagnetic transition. To minimize the effects due to inter-domain interfaces, we consider large scale simulations comprising $N = 8000$ particles, with $N_1 = N_2$. The gas parameter is $na^3 = 10^{-4}$. Fig. 4 shows the position snapshots observed after thermalization is reached. Two initial particle configurations are considered. They feature either vertically separated or mixed components. In the separated configuration, the first component is uniformly randomly distributed in the lower half of the 3D simulation box, while the second component is in the upper half. In the mixed initial configuration, both components are uniformly distributed in the whole box. In panels (a) and (b), the interspecies coupling strength is $g_{12}/g = 0.93$, i.e., below the $T = 0$ MF critical point. In the first panel, the temperature is relatively low, namely, $T/T_c^0 \cong 0.238$. Here, even Popov theory would predict a paramagnetic state. In the second, it is closer to the BEC transition temperature, where Popov theory would predict a ferromagnetic state. The two simulations start in the separated configuration. Despite of being initially separated, the two components rapidly mix, indicating a paramagnetic

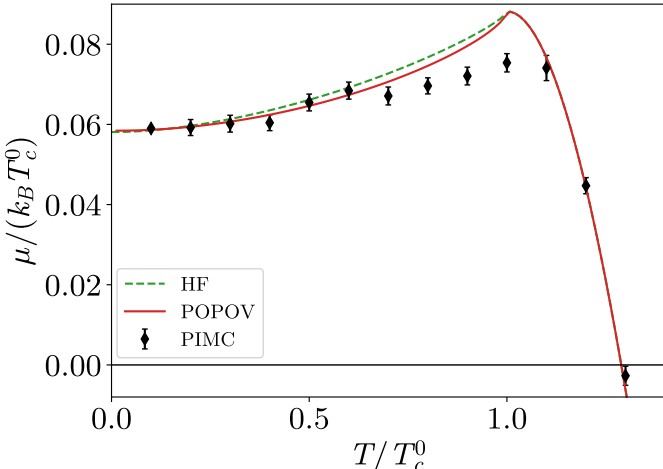

Figure 5: Chemical potential of an unpolarized mixture with $na^3 = 10^{-6}$, and $g_{12}/g = 0.93$ as a function of the temperature. The PIMC results in the thermodynamic limit (black points) are compared with the HF (green dashed line) and the Popov (red solid line) predictions.

state, both at low temperature and closer to the BEC transition. In panel (c), the inter-species interaction strength ($g_{12}/g = 1.2$) is beyond the critical point predicted by the MF theory. In this case, the two components keep the initial spatial separation along the vertical direction, with only minor mixing close to the interface separating the two domains. Interestingly, even when they start from a mixed configuration [panel (d)], they form two well defined ferromagnetic domains. This indicates that large-scale PIMC simulations are able to simulate phase separated states. Chiefly, these observations further corroborate the claim that the finite temperature transition corresponds to the $T = 0$ MF scenario, in contrast to the HF and Popov predictions. When the temperature is raised [panel (e)], the interface is less regular and it looses memory of the initial position. One also notices a larger impurity density, corresponding to a ferromagnetic state with partial polarization. Moving even closer to the BEC transition temperature $T_c^0$ [panel (f)], the two domains can be hardly identified by naked eye. However, we argue that in the thermodynamic limit one would still observe a (partially) ferromagnetic state, meaning that the Curie critical temperature where melting occurs is even higher.

## 4 Thermodynamic properties of balanced mixtures

We now turn our attention to the study of thermodynamic quantities, focusing on the $g_{12} = 0.93g$ sodium mixture in the balanced state $p = 0$. In this case we have chosen the value $na^3 = 10^{-6}$ for the gas parameter which is closer to experimentally relevant conditions in the absence of Feshbach resonances. The PIMC results for the thermodynamic quantities shown below are the extrapolations to the thermodynamic limit of the data computed with up to 512 total particles. In Fig. 5 we show the chemical potential $\mu = \mu_1 = \mu_2$ of the mixture as a function of temperature below and above the transition point and we compare it with the results of perturbative approaches. The results are in good agreement with both HF and Popov predictions when the temperature is not too close to the critical point. In the critical region around $T_c^0$, deviations are sizable. They tend to suppress the maximum, similarly to the results of Fig. 1 for the minority component. We notice that a maximum in the temperature dependence of the chemical potential should be expected on general grounds from the theory

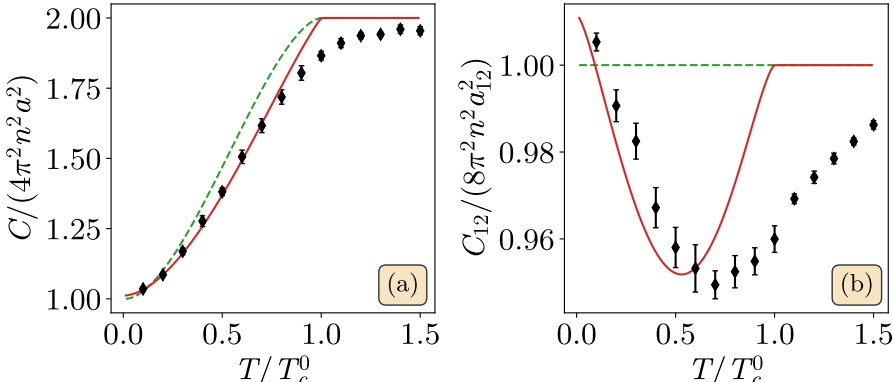

Figure 6: Intraspecies (*panel* (a)) and interspecies (*panel* (b)) contact parameters of an unpolarized mixture with $na^3 = 10^{-6}$, and $g_{12}/g = 0.93$ as a function of the temperature. The PIMC results in the thermodynamic limit (black points) are compared with the HF (green dashed line) and the Popov (red solid line) predictions.

of superfluids and has been recently observed in a single-component dilute Bose gas [27]. The PIMC results for $\mu$ in a single-component gas are discussed in the appendix as a test study of the chemical potential algorithm.

In Fig. 6 we show the results for the contact parameters, important thermodynamic quantities sensitive to short-range correlations. In a symmetric unpolarized mixture one defines two contact parameters $C_{11} = C_{22} = C$ and $C_{12}$ associated to correlations within each component and between the two components respectively

$$C = 16\pi^2 a^2 \frac{\partial F/V}{\partial g}, \qquad C_{12} = 32\pi^2 a_{12}^2 \frac{\partial F/V}{\partial g_{12}}. \qquad (2)$$

The contact parameter $C$ has been measured as a function of temperature in a single-component gas [28] and in a mixture of a Bose gas with impurities [29]. In our PIMC simulations we have computed $C$ and $C_{12}$ from the short-range behavior of the pair correlation function for particles belonging to the same and to different components [21]. The results for $C$ are in good agreement with both HF and Popov predictions, showing deviations only in the vicinity of $T_c^0$. For $C_{12}$, instead, the HF prediction does not depend on the temperature, while the Popov prediction yields a small minimum. Our PIMC findings show a small minimum around $T \simeq 0.7 T_c^0$ which reproduces this. This enhanced repulsive correlation between the two components at intermediate temperatures has been already discussed in repulsive mixtures [30, 31] and deserves further investigation.

# 5 Conclusion

We have investigated the magnetic and thermodynamic properties of repulsive Bose mixture using exact numerical methods. For the values of the parameters considered in the simulations we do not find the ferromagnetic transition predicted to occur at finite temperature by perturbative approaches and we find good agreement with the magnetic susceptibility from simple mean-field theory at zero temperature. We further argue that a similar conclusion is expected to hold for lower values of the gas parameter. This claim is further corroborated by the analysis of particle-positions snapshots. Thermodynamic quantities reveal the role of critical fluctuations close to the BEC transition point and the behavior of the contact parameters contains important information on short-range correlations in the mixture that can be mea-

sured in future experiments. Our findings indicate the importance of unbiased simulations for atomic mixtures, in contrast to previous perturbative treatments of repulsive and attractive two-component Bose gases.

## Acknowledgments

**Funding information** This work was supported by the Italian Ministry of University and Research under the PRIN2017 project CEnTraL 20172H2SC4. S.P. acknowledges support from the PNRR MUR project PE0000023-NQSTI, and PRACE for awarding access to the Fenix Infrastructure resources at Cineca, which are partially funded from the European Union's Horizon 2020 research and innovation programme through the ICEI project under the grant agreement No. 800858. J.B. and G. P. acknowledge financial support from Ministerio de Economia, Industria y Competitividad (MINECO, Spain) under grant No. PID2020-113565GB-C21. L.P, N.G.P and T.P.B acknowledge support from the UK Engineering and Physical Sciences Research Council (Grant No. EP/T015241/1).

## A  PIMC computation of chemical potential and free energy

In this appendix, we present the details of the PIMC algorithm we employ for the computation of the chemical potential of a Bose gas. The basic idea is to recognize that the chemical potential can be derived from the ratio of the partition functions for the systems with $N + 1$ and $N$ particles (at fixed volume and temperature) as

$$\mu(N, T) = F(N + 1, T) - F(N, T) = -k_B T \log \frac{Z_{N+1}}{Z_N}. \tag{A.1}$$

As noted in Ref. [23] the above formula can be leveraged in a canonical PIMC calculation by enlarging the configurational space to include the sector with one additional particle. The ratio $Z_{N+1}/Z_N$ is then evaluated as the relative time spent by the simulation in the two sectors. The simulation resembles a grand canonical one, with the difference that it is restricted to states with either $N$ or $N+1$ particles, thus providing higher statistics for the computation of $\mu(N, T)$. Combining the chemical potential with the pressure, we can obtain the free energy

$$F = \Omega + \mu N = -PV + \mu N, \tag{A.2}$$

where $\Omega = -PV$ is the grand canonical potential.

### A.1  Details of the algorithm

In order to extend the algorithm described in Ref. [22] and enable the computation of the chemical potential we work with $N + 1$ polymers and implement a boolean variable for each polymer to activate or deactivate it.

The configurational space is now composed by four sectors: the original $Z_N$ and $G_N$ together with the corresponding sectors with one additional particle $Z_{N+1}$ and $G_{N+1}$ and one needs to introduce appropriate Monte Carlo moves to allow the Markov-Chain to visit all the configurations within these sectors. The four sectors together with the sector-changing moves are summarized in Fig. 7. In general one can introduce a grand canonical chemical potential $\mu_{\text{gc}}$ as a simulation parameter to be used to increase the sampling efficiency. In particular it can be tuned to be close to the expected value e.g. by using the Hartree-Fock approximation,

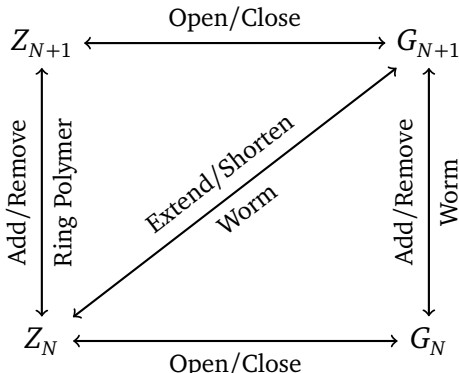

Figure 7: The four sectors interconnected by the web of sector-changing moves.

in order to balance the simulation time spent within the sectors with $N$ and $N+1$ particles. The chemical potential is then evaluated as

$$\mu(N,T) = \mu_{\text{gc}} - k_B T \log \frac{t(Z_{N+1})}{t(Z_N)}, \tag{A.3}$$

where $t(Z_{N+1})/t(Z_N)$ is the ratio of the simulation time spent in the two sectors $Z_{N+1}$ and $Z_N$. We have implemented three sets of particle-number changing moves—*Extend/Shorten Worm*, *Add/Remove Worm* and *Add/Remove Ring Polymer*—that are briefly described below using the notation of Ref. [22] and indicating with $\Delta U$ the variation in the potential energy between the new proposed configuration and the old one. Within the primitive approximation we would have $\Delta U = \frac{\beta}{M} \sum_j \left( V_j' - V_j \right)$, where $M$ is the total number of imaginary-time slices and $V_j'$ ($V_j$) is the sum of the two-body potentials over all pairs of particles at the slice $j$ after (before) the Monte Carlo update. Note that, when in the sectors with $N$ particles, one must be careful to exclude the deactivated polymer from the computation.

**Extend/Shorten Worm**    These moves connect the sectors $G_N$ and $G_{N+1}$ by adding or removing a polymer at the end of the worm. To extend the worm we first check if sector is $G_N$, then we activate the extra polymer and we put it in permutation with the worm's head. We then use the staging algorithm to redraw the last polymer as in *Move Head*. The move is accepted with probability

$$A_{EX} = \min \left\{ 1, e^{\beta \mu_{\text{gc}} - \Delta U} \right\}. \tag{A.4}$$

To shorten the worm we first check if sector is $G_{N+1}$ and if the worm is at least two polymers long. Then we deactivate the last polymer of the worm. The move is accepted with probability

$$A_{SH} = \min \left\{ 1, e^{-\beta \mu_{\text{gc}} - \Delta U} \right\}. \tag{A.5}$$

**Add/Remove Worm**    These moves connect the sectors $Z_N$ and $G_{N+1}$ by adding or removing a one-polymer worm. To add the worm we first check if sector is $Z_N$, then we activate the extra polymer, we uniformly sample its first bead in the volume and we use the staging algorithm to sample the rest of the polymer as in *Move Head*. The move is accepted with probability

$$A_{AW} = \min \left\{ 1, C e^{\beta \mu_{\text{gc}} - \Delta U} \right\}, \tag{A.6}$$

where $C$ is the open/close parameter. The complementary move consists in removing a one-polymer long worm from the $G_{N+1}$ sector by deactivating it. The move is accepted with probability

$$A_{RW} = \min \left\{ 1, C^{-1} e^{-\beta \mu_{\text{gc}} - \Delta U} \right\}. \tag{A.7}$$

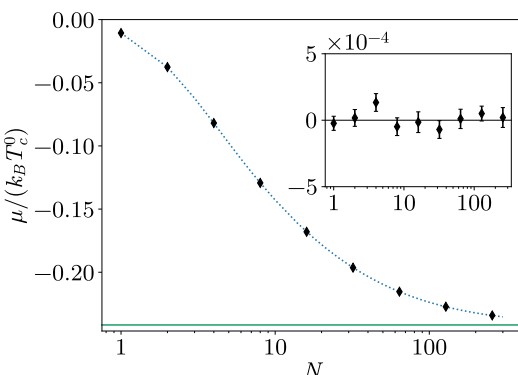

Figure 8: Chemical potential of an ideal Bose gas with $N$ particles at temperature $T = 1.5T_c^0$. The PIMC values (black diamonds) are compared with the exact results at fixed $N$ (connected by the blue dotted line) and with the result in the thermodynamic limit (green horizontal line). *Inset*: the difference between the PIMC and the exact values.

**Add/Remove Ring Polymer**   These moves connect the sectors $Z_N$ and $Z_{N+1}$ by adding or removing a ring polymer (i.e. a polymer in permutation with itself and with zero winding). To add the ring we first check if sector is $Z_N$, then we activate the extra polymer and we uniformly sample its first bead in the volume. The last bead $M$ of the polymer is then set to be equal to the first and we use the staging algorithm to sample the rest of the polymer. The move is accepted with probability

$$A_{AR} = \min\left\{1, \frac{V}{(N+1)\lambda_T^D} e^{\beta\mu_{\mathrm{gc}} - \Delta U}\right\}. \tag{A.8}$$

The complementary move consists in removing a one-polymer ring with zero winding from the $Z_{N+1}$ sector by deactivating it. The move is accepted with probability

$$A_{RR} = \min\left\{1, \frac{(N+1)\lambda_T^D}{V} e^{-\beta\mu_{\mathrm{gc}} - \Delta U}\right\}. \tag{A.9}$$

## A.2   Benchmarks

Following the strategy of Ref. [22] we carefully check our implementation by running a number of tests. First of all, we verify that we correctly recover the values of the chemical potential for the ideal Bose gas for each system size $N$. In Fig. 8 we show the PIMC results at the temperature $T = 1.5T_c^0$ compared with the exact values (obtained via the recursion formulas as in Refs. [32,33] and reviewed in Ref. [22]) and with the result in the thermodynamic limit given by

$$\mu_{\mathrm{IBG}} = k_B T \log(z), \tag{A.10}$$

where $z$ is an effective fugacity that determines the total density of the gas via the equation $n\lambda_T^3 = g_{3/2}(z)$ with $g_\nu(z)$ the usual special Bose functions. The agreement between the PIMC data and the expected values is perfect at any size and does not depend on the number of imaginary-time slices used in the simulation. Moreover we verify that below $T_c^0$ the PIMC results are compatible with a zero chemical potential.

We then benchmark the interacting gas, where the repulsive interaction is modeled by a hard sphere potential. As in Ref. [21], we use the pair-product ansatz [34] for the computation of the potential energy $\Delta U$. In Fig. 9 we compare the PIMC results for the chem-



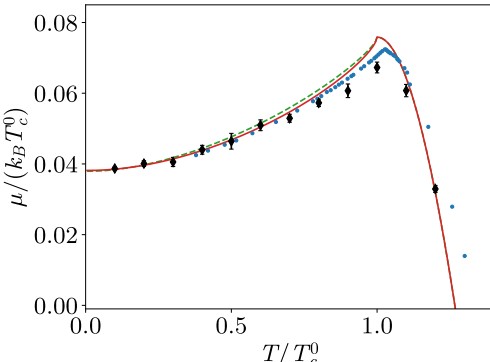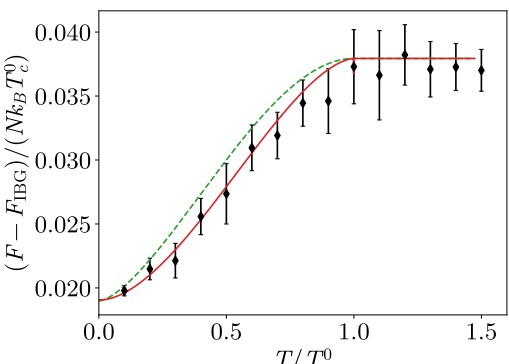

Figure 9: Results for an interacting Bose gas with gas parameter $na^3 = 10^{-6}$ as a function of the temperature. The PIMC values, extrapolated to the thermodynamic limit (black crosses), are compared with the perturbative results of Hartree-Fock (green dashed line) and Popov (red solid line) theories. *Left panel:* Results for the chemical potential, also compared with the predictions from the universal relations of Ref. [26] (blue dots). *Right panel:* difference in freee energy with the ideal Bose gas result.

ical potential and the free energy with the perturbative predictions. The PIMC data are extrapolated to the thermodynamic limit using a linear fit in $1/N$ of the results for four sizes $N = 128, 256, 384, 512$. The number of imaginary-time slices is 16 for all sizes. For the chemical potential, we also compare our results with the predictions from the universal relations of Ref. [26], using their data for the density shift $\lambda(X) \propto n - n_c$ to extract the reduced temperature $t = T/T_c^0$ and then mapping it to the corresponding chemical potential using the data for the chemical potential shift $X \propto \mu - \mu_c$. In particular, expressing the relations in our units, we find that each value of $\lambda(X)$ can be mapped to a value of $t$ solving the equation

$$\frac{16\pi^3}{\zeta(3/2)} a n^{1/3} (\lambda(X) - \mathcal{C}) t^2 + t^{3/2} = 1, \tag{A.11}$$

in the region where the universal relations can be applied, namely $t \sim 1$. The numerical constant $\mathcal{C}$ is determined as $\mathcal{C} = 0.0142(4)$. From the corresponding values of $X$ we then determine the chemical potential shift as

$$\frac{\mu - \mu_c}{k_B T_c^0} = \frac{32\pi^3 a^2 n^{2/3}}{\zeta(3/2)^{2/3}} t^2 X. \tag{A.12}$$

Finally, to get the sought-after values of $\mu$, we need to add the values of $\mu_c$ as obtained from Ref. [35]

$$\frac{\mu_c}{k_B T_c^0} = 4 a n^{1/3} \zeta(3/2)^{2/3} t^{3/2} - \frac{32\pi a^2 n^{2/3}}{\zeta(3/2)^{2/3}} t^2 \log\left( \mathcal{K} \frac{\zeta(3/2)^{1/3}}{a n^{1/3} \sqrt{32\pi^3 t}} \right), \tag{A.13}$$

where $\mathcal{K} = 0.673(1)$ is a numerical constant.[1] The data for $\mu$ obtained from the universal are represented by the blue dots in the left panel of Fig. 9 and show a good agreement with the PIMC data for $T < T_c^0$, while, for $T > T_c^0$, a discrepancy builds up for increasing temperatures, since the universal relations are valid only in the regime of large occupation numbers for single-particle modes. In that regime, the PIMC data nicely reproduce the HF predictions. With the benchmarks shown so far, we are now confident that the PIMC algorithm correctly reproduces the physics of a single-component Bose gas both in the non-interacting and in the interacting case. In the following section we show how to extend the algorithm for Bose mixtures.

---

[1]We warn the reader that the expression for $\mu_c$ that can be found in Ref. [26] is not correct. We thank the authors for clarifying the issue.

# B  PIMC algorithm for a binary Bose mixture

Extending the PIMC algorithm to the case of multicomponent gases is pretty straightforward: one just needs to restrict the *Swap* move to involve only particles of the same species and, in the interacting case, to take into account the inter-species interaction described by the *s*-wave scattering length $a_{12}$. The computation of the chemical potentials for the two species proceeds as before via the free energy difference, this time making sure the number of particles of the other species is kept fixed. For a two-component mixtures we have:

$$\mu_1(N_1, N_2, T) = F(N_1 + 1, N_2, T) - F(N_1, N_2, T),$$
$$\mu_2(N_1, N_2, T) = F(N_1, N_2 + 1, T) - F(N_1, N_2, T). \tag{B.1}$$

where the differences are numerically computed as the ratios of Monte Carlo times spent in the different sectors. The simulation now lives in a configurational space made by $4 \times 4 = 16$ sectors. Several consistency checks where made on the algorithm. In Fig. 10 we show the results for two non-interacting ideal Bose gases, where we recover the known exact result as a function of the polarization. The chemical potential $\mu_1$ for the majority component is consistent with zero, while the chemical potential $\mu_2$ for the minority component becomes non-zero above the critical polarization, where it becomes normal.

Using the above method for computing the chemical potentials, one can obtain the value of the free energy of the mixture via the thermodynamic relation

$$F = -PV + \mu_1 N_1 + \mu_2 N_2. \tag{B.2}$$

Notice that, while this quantity contains valuable information and it represents our main test-bench for the perturbative predictions, it comes at the cost of a cancellation between the pressure term and the chemical potential terms, that amplifies its final statistical error. However, when we focus on the magnetic properties of a binary mixture, we are only interested in the *free energy difference* among mixtures at different values of the polarization $p = (N_1 - N_2)/N$, where $N = N_1 + N_2$ is the total number of particles. Such a difference can be evaluated more efficiently by devising an algorithm that directly samples configurations with different values of the polarization, while keeping $N$ fixed. Denoting with $Z_{N,p}$ the partition function with $N$ total particles and polarization $p$, the free energy difference $\Delta F(N, p)$ between the state at polarization $p$ and the unpolarized state with $p = 0$ can be computed as

$$\Delta F(N, p) = F\left(\frac{N(1+p)}{2}, \frac{N(1-p)}{2}, T\right) - F\left(\frac{N}{2}, \frac{N}{2}, T\right) = -k_B T \log \frac{t(Z_{N,P})}{t(Z_{N,0})}, \tag{B.3}$$

where the ratio $t(Z_{N,P})/t(Z_{N,0})$ is the ratio between the time spent in the sector with polarization $p$ and the time spent in the sector with zero polarization. There are many possible ways to implement such an algorithm: One possibility is, for example, to combine the moves of Sec. A.1 for the two species in such a way that each time a particle of one species is created a particle of the other species is removed. In the following we mention another possibility, which is slightly more sophisticated.

## B.1  Details of the algorithm for free energy differences in a mixture

An efficient algorithm that spans the configurations with different polarizations, while keeping $N$ fixed can be devised by taking close inspiration from the original grand canonical implementation of Refs. [20, 36]. The Monte Carlo moves have been adapted in such a way that both the total number of polymers and the total number of beads are kept constant throughout the simulation. Within this algorithm, the worms for the two species might be present simultaneously, also in a configuration where the beads of one polymer are shared between the two

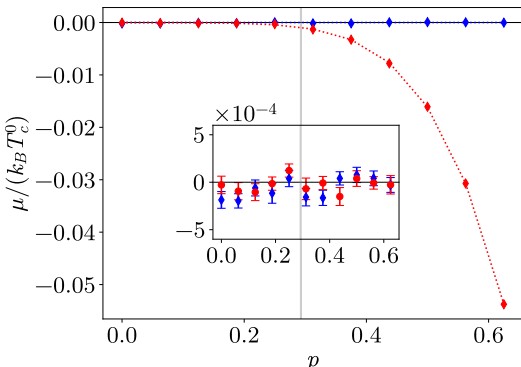

Figure 10: Chemical potentials $\mu_1$ (blue) and $\mu_2$ (red) for a non-interacting mixture of ideal Bose gases as a function of the polarization. The temperature is kept fixed at $T = 0.5T_c^0$ and the total number of particles is $N = 128$. The PIMC points are compared to the exact results connected by the dotted lines. The critical polarization, at which the minority component becomes normal, is signaled by a gray vertical line. *Inset*: the difference between the PIMC data and the exact results.

worms. For example, the polymer $i_0$ might be filled by the species 1 up to the imaginary-time slice $j_0$ (corresponding to the head of the worm 1), while the rest of the slices are filled by the worm of the species 2 (that has its tail at the imaginary-time slice $j_0 + 1$ of the polymer $i_0$). We briefly describe below a minimal pair of moves, called *Advance/Recede*, that allows the simulation to span the configurations at different values of polarization (except the case at $p = 1$). Other moves can be included in order to improve the ergodicity of the Markov chain across the sectors, for example by combining *Advance/Recede* with *Open/Close*. The details of these combined moves will not be given here; instead we just outline the aforementioned minimal addition that can be used for small values of the polarization.

The *Advance* and *Recede* moves change the relative lengths of the worms and can only be performed when both worms are present. In the *Advance* move the head of the worm of species $s$ is advanced in imaginary time from the slice $j$ to the slice $j + \Delta j$, by sampling the new $\Delta j$ beads with the staging algorithm as in *Move Head*. The tail of the worm of the other species $s'$ is advanced as well by deleting all the beads between the slice $j$ and the slice $j + \Delta j$. Note that we must reject the move if the worm of species $s'$ is completely deleted by the proposed update. The move is then accepted with probability

$$A_{\text{advance}} = \min\left\{1, e^{\beta \Delta \mu \, \Delta j / M - \Delta U}\right\}, \tag{B.4}$$

where $\Delta \mu = \mu_{\text{gc}}^s - \mu_{\text{gc}}^{s'}$ is the difference between the grand canonical chemical potentials of the two species. The complementary *Recede* move is completely symmetric and can be obtained as the *Advance* move with negative values of $\Delta j$. It consists in receding the head of the species $s$ by deleting $\Delta j$ beads, while simultaneously creating $\Delta j$ new beads for the species $s'$. The move is accepted with probability

$$A_{\text{recede}} = \min\left\{1, e^{-\beta \Delta \mu \, \Delta j / M - \Delta U}\right\}. \tag{B.5}$$

This pair of moves can change the species of whole polymers, thus allowing the algorithm to sample configurations with different polarizations. We have checked that the free energy differences computed directly through Eq. (B.3) reproduce those obtained from the full computation of the free energy, but deliver smaller statistical errors.

# C  Hartree-Fock and Popov theories

The Hartree-Fock and Popov theories of repulsive binary Bose mixtures at finite temperature are described in details in Refs. [15, 16]. We note that Popov's theory is also known as the finite temperature extension of Beliaev's approach and includes the important contribution of anomalous fluctuations to thermodynamic quantities [37, 38]. Here we report the results for the Helmholtz free energy obtained in the two approaches from which all thermodynamic quantities discussed in the main text can be derived.

Within the HF approximation one finds

$$\frac{F_{\mathrm{HF}}}{V} = \frac{g}{2}\left(n_1^2 + n_2^2\right) + g_{12}n_1n_2 + gn_T^{0\,2} + \frac{1}{\beta V}\sum_{\mathbf{k}}\left[\ln\left(1 - e^{-\beta(\epsilon_k + gn_{1,0})}\right) + \ln\left(1 - e^{-\beta(\epsilon_k + gn_{2,0})}\right)\right],$$

(C.1)

holding when both condensates are present, *i.e.* in the polarization range $p < p_c$ set by the critical polarization $p_c = 1 - (T/T_c^0)^{3/2}$ at which the minority component 2 turns normal. Here $\epsilon_k = \hbar^2 k^2/(2m)$ is the single-particle kinetic energy and $n_T^0 = \zeta(3/2)/\lambda_T^3$ is the non-interacting thermal density written in terms of the thermal wavelength $\lambda_T = \sqrt{2\pi\hbar^2/mk_BT}$ and $\zeta(3/2) \simeq 2.612$. Furthermore, $n_{i,0}$ ($i = 1, 2$) correspond to the condensate density of the two components calculated to lowest order in the interaction strength: $n_{i,0} = n_i - n_T^0$. When $p > p_c$ and the density $n_2$ of the minority component does not exceed the thermal density $n_T^0$, the above expression for free energy becomes

$$\frac{F_{\mathrm{HF}}}{V} = \frac{g}{2}\left(n_1^2 + 2n_2^2 + n_T^{0\,2}\right) + g_{12}n_1n_2 + \mu_2^{\mathrm{IBG}}n_2$$
$$+ \frac{1}{\beta V}\sum_{\mathbf{k}}\left[\ln\left(1 - e^{-\beta(\epsilon_k + g(n_1 - n_T^0))}\right) + \ln\left(1 - e^{-\beta(\epsilon_k - \mu_2^{\mathrm{IBG}})}\right)\right],$$

(C.2)

where the effective chemical potential $\mu_2^{\mathrm{IBG}}$ is fixed by the normalization condition of the minority component $n_2 = g_{3/2}(e^{\beta\mu_2^{\mathrm{IBG}}})/\lambda_T^3$, with $g_{3/2}(z)$ the usual special Bose function. Notice that expressions (C.1) and (C.2) coincide at $p = p_c$ where $n_2 = n_T^0$ and $\mu_2^{\mathrm{IBG}} = 0$.

The Popov theory includes the contribution from collective excitations (density and spin waves) into the thermodynamics of the mixture yielding the following expression for the free energy:

$$\frac{F}{V} = \frac{g}{2}\left(n_1^2 + n_2^2\right) + g_{12}n_1n_2 + gn_T^{0\,2}$$
$$+ \frac{1}{\beta V}\sum_{\pm}\sum_{\mathbf{k}}\ln\left(1 - e^{-\beta E_k^{\pm}}\right) + \left(\frac{m}{2\pi\hbar^2}\right)^{3/2}\frac{4}{15\sqrt{\pi}}\sum_{\pm}(2\Lambda_{\pm})^{5/2},$$

(C.3)

valid when both components are in the condensed state ($p < p_c$). The first term in the second line collects the thermal contribution from the excitation spectrum in the density and spin channel $E_k^{\pm} = \sqrt{\epsilon_k^2 + 2\Lambda_{\pm}\epsilon_k}$ whereas the last term survives also at $T = 0$ yielding the Lee-Huang-Yang beyond mean-field corrections to the ground-state energy. Both terms involve the effective chemical potentials

$$\Lambda_{\pm} = \frac{1}{2}\left(gn_0 \pm \sqrt{(g^2 - g_{12}^2)n^2p^2 + g_{12}^2 n_0^2}\right),$$

(C.4)

where $n_0 = n - 2n_T^0$ is the condensate density calculated to lowest order in the interaction

strength. In the regime of high polarization ($p > p_c$) the above expression reduces to

$$
\begin{aligned}
\frac{F}{V} = {} & \frac{g}{2}\left(n_1^2 + 2n_2^2 + n_T^{0\,2}\right) + g_{12}n_1 n_2 \\
& + \left(\frac{m}{2\pi\hbar^2}\right)^{3/2}\frac{4}{15\sqrt{\pi}}\left(2g(n_1 - n_T^0)\right)^{5/2} + \mu_2^{\mathrm{IBG}}n_2 \\
& + \frac{1}{\beta V}\sum_{\mathbf{k}}\left[\ln\left(1 - e^{-\beta\sqrt{\epsilon_k^2 + 2\epsilon_k g(n_1 - n_T^0)}}\right) + \ln\left(1 - e^{-\beta(\epsilon_k - \mu_2^{\mathrm{IBG}})}\right)\right],
\end{aligned}
\tag{C.5}
$$

where similarly to the HF case the effective chemical potential $\mu_2^{\mathrm{IBG}}$ is determined by the normalization condition $n_2 = g_{3/2}(e^{\beta\mu_2^{\mathrm{IBG}}})/\lambda_T^3$.

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
