# Peer review of "Phase separation in binary Bose mixtures at finite temperature"

_SciPost Physics, doi:SciPost Phys. 15, 171 (2023)_

## Round 1 · Referee Report · Anonymous (Referee 1) · 2023-5-1

Report
In their manuscript, the authors study the magnetic behavior of finite-temperature repulsive two-component Bose mixtures using exact path-integral Monte-Carlo simulations. The authors implement a new algorithms for the free energy, the chemical potential, and the magnetic susceptibility of the two components. The main result is that conditions for phase separation remain unchanged from the zero temperature case. The calculation of the chemical potential and contact parameters as a function of temperature, show their deviations in the critical region from the results of perturbative methods.
Overall, the manuscript is well presented, and can be considered as a timely contribution to an interesting topic. However there are some mysterious points that should be addressed more carefully.
The authors don’t provide much of a physical explanation for their results. For example, the statement « the conditions for phase separation are not modified from the zero temperature case ». The choice of the parameters : N=128 and T=0.794 T_c. Is this choice arbitrary ? N looks small, what happens if one extends it to large values?
The authors compared their Monte Carlo data for the chemical potential and free energy only with their own HF and Popov theories but they completely ignored the other theories such as the full HFB theory (see e.g. Phys. Rev. A 97, 033627 (2018) and Phys. Rev. A 104, 023310 (2021). The discrepancy between the Monte Carlo simulations and the HF and Popov theories couldn’t due to the missing of the anomalous correlations (pairing) in these perturbative theories? Effects of thermal fluctuations on the phase separation in Bose mixtures has been also discussed in Phys. Rev. A 97, 033627 (2018). The above references should be cited and commented on.
In conclusion, before giving my definitive approval, I recommend that the authors provide more details into the physical reasons behind the obtained results. I think this would be of great benefit not only to the manuscript and also to the reader.
Overall, the manuscript is well presented, and can be considered as a timely contribution to an interesting topic. However there are some mysterious points that should be addressed more carefully.
The authors don’t provide much of a physical explanation for their results. For example, the statement « the conditions for phase separation are not modified from the zero temperature case ». The choice of the parameters : N=128 and T=0.794 T_c. Is this choice arbitrary ? N looks small, what happens if one extends it to large values?
The authors compared their Monte Carlo data for the chemical potential and free energy only with their own HF and Popov theories but they completely ignored the other theories such as the full HFB theory (see e.g. Phys. Rev. A 97, 033627 (2018) and Phys. Rev. A 104, 023310 (2021). The discrepancy between the Monte Carlo simulations and the HF and Popov theories couldn’t due to the missing of the anomalous correlations (pairing) in these perturbative theories? Effects of thermal fluctuations on the phase separation in Bose mixtures has been also discussed in Phys. Rev. A 97, 033627 (2018). The above references should be cited and commented on.
In conclusion, before giving my definitive approval, I recommend that the authors provide more details into the physical reasons behind the obtained results. I think this would be of great benefit not only to the manuscript and also to the reader.

Author: Sebastiano Pilati on 2023-06-22 [id 3753]
(in reply to Report 1 on 2023-05-01)We thank the Referee for the positive assessment on the presentation, relevance, and timeliness of our manuscript. In the following and in the revised manuscript we answer the specific points raised by the them, providing the requested clarifications on our findings and on the adopted approach.
The referee writes:
Our response: This statement is based on the results shown in Fig.2 and in Fig.3. In fact, for all values of g_{12}<g, the dependence of the free energy on the polarization p is in very good agreement with the mean-field T=0 prediction for the magnetic susceptibility [see dotted blue line in panels a) and b) of Fig.2 and in Fig.3] and does not show any evidence of the ferromagnetic transition predicted by theories including beyond mean-field effects. Only when g_{12}>g [see panel d) in Fig.2] a minimum in free energy shows up at finite p, meaning that the mixture turns ferromagnetic. This is exactly the behavior expected at T=0, and it is recovered here at a high temperature not so far from the BEC transition point. Our conclusion is that we expect the same to be true also for lower temperatures, where thermal effects not captured by the mean-field description should play a minor role. In this respect one should also notice that higher order interaction effects at T=0 do not change the critical value g_{12}=g for the onset of ferromagnetism (see Ref.[16]). As an additional remark, we point out that our results do not exclude a non trivial interplay between ferromagnetic and critical fluctuations in the close vicinity of the transition point. Such studies would require a much deeper analysis of the shift of the transition point in interacting mixtures, which is clearly beyond the scope of this work. To better clarify the implications drawn from our results we have rewritten the paragraph at the end of the section where we discuss Fig.2 and Fig.3. For convenience, the new paragraph is reported hereafter: “From these results we conclude that, in contrast to HF and Popov predictions, the magnetic susceptibility depends very little on the temperature, and the conditions for phase separation seem to remain the same as at $T=0$. In fact, if $g_{12}<g$, our results indicate that the only thermodynamically stable phase is the paramagnetic state at $p=0$. A ferromagnetic state forms when $g_{12}>g$ and the effect of temperature is to reduce the equilibrium polarization from the $p=1$ value achieved only at zero temperature. This is found at a high temperature not far from the BEC transition point and we expect the same to be true also for lower temperatures, where thermal effects not captured by the mean-field description should play a minor role. In this respect one should also notice that higher order interaction effects at T=0 do not change the critical value g_{12}=g for the onset of ferromagnetism (see Ref.[16]). As an additional remark, we point out that our results do not exclude a non trivial interplay between ferromagnetic and critical fluctuations in the close vicinity of the transition point. To carefully investigate these effects would require a much deeper analysis of the shift of the transition point in interacting mixtures beyond the scope of this work. Furthermore, we expect the simple $T=0$ scenario to hold also at densities lower than $na^3=10^{-4}$. We checked…”
The referee writes:
Our response: Indeed we mostly focus our analysis on the temperature T=0.794 T_c (we also consider different temperatures in the snapshots of particle configurations shown in Fig.4). However, the relevant behavior to establish the magnetic response concerns the polarization p at fixed temperature. The latter should be high enough to emphasize thermal effects, but not to close to the transition point to make sure that the mixture is still in the Bose condensed phase. Concerning the number of particles, we have checked that our results for the free energy do not change significantly by increasing N and can be considered well converged approaching the thermodynamic limit. For clarity we added two comments on the choice of the temperature and on the role of finite-size effects. For convenience, these two comments are reported hereafter: “This choice of parameters and, in particular, the choice of temperature emphasizes thermal effects in HF and Popov theories yielding important corrections to the $T=0$ magnetic susceptibility. We also note that finite-size effects in PIMC simulations of the free energy are negligible if one increases further the total number of particles.”
The referee writes:
Our response: We thank the Referee for pointing out a possible misunderstanding on the content of what we refer to as HF and Popov theories, and for suggesting relevant references. The Popov theory we use here, which is explicitly outlined in Appendix C, consistently accounts for second order effects in the coupling constant. It is also known as second-order Beliaev theory extended to finite temperatures. In particular, at T=0, it provides the correct Lee-Huang-Yang terms in the equation of state for the single component and the similar terms arising from the zero-point motion of density and spin fluctuations in two-component mixtures. In the language of the Referee it accounts properly for both normal and anomalous correlations yielding at finite temperature a description of the thermodynamic behavior which extends to mixtures Popov’s result for the single-component gas (see Ref.[16]). The only difference compared to full HFB theory is that in the dispersion of elementary excitations at finite temperature we use the condensate density as calculated from lowest order theory instead of self-consistently. However, this approach is consistent up to second-order corrections and should be adequate in the dilute regime. We have tried to address this issue adding a clarifying comment at the beginning of Appendix C and making reference to the useful article pointed to by the Referee. For convenience, the additional comment and references are reported hereafter: “The Hartree-Fock and Popov theories of repulsive binary Bose mixtures at finite temperature are described in details in Refs.~\cite{PhysRevLett.123.075301, PhysRevA.102.063303}. We note that Popov’s theory is also known as the finite temperature extension of Beliaev’s approach and includes the important contribution of anomalous fluctuations to thermodynamic quantities \cite{Phys. Rev. A 97, 033627 (2018); Phys. Rev. A 104, 023310 (2021)}. Here we report…”
Anonymous on 2023-07-04 [id 3778]
(in reply to Sebastiano Pilati on 2023-06-22 [id 3753])In this revised version, the authors have improved the manuscript with
respect to the previously submitted version. They answer to almost of my comments. I therefore recommend acceptance of this manuscript for publication in SciPost.

---

## Round 2 · Referee Report · Anonymous · 2023-7-4

Report

In this revised version, the authors have improved the manuscript with respect to the previously submitted version. They answer to almost of my comments. I therefore recommend acceptance of this manuscript for publication in SciPost.

---

## Round 2 · Referee Report · Anonymous · 2023-8-28

Strengths

1-Long standing questions are probed, such as the validity of certain perturbative theories, and the question of whether the ferromagnetic transition can be traversed by varying temperature.

2- the results provide an important and interesting contribution to better understanding the role of critical fluctuations on the properties of bose-bose mixtures

3-The manuscript is well written.

Weaknesses

1- focus is given to only a limited range of temperature and interaction strengths

2-The error bars on the Monte Carlo results are relatively large

Report

This manuscript investigates phase separation and thermodynamic properties of Bose mixtures using path integral Monte Carlo simulations. The focus is on the effects of temperature, and to what extent Hartree Fock and Popov theories are inaccurate in the vicinity of the BEC phase transition.

By calculating the free energy and chemical potentials they find ferromagnetic phases for which the minority component is in the normal phase, suggesting that in this regime, lowering the temperature would transition to the zero temperature prediction for a paramagnetic phase. A main result is that in this regime the Monte Carlo theory does not predict this same temperature-driven ferromagnetic transition. Also, near the transition there are significant shortcomings of the HF and Popov theories.

These results are interesting and provide an important piece for a long standing puzzle, which has continued to be of interest over the years. For this reason I would be in favor of supporting publication if the authors are able to address my technical concerns/questions below.

Requested changes

1-Thermal fluctuations are expected to diverge leading up to the ferromagnetic transition, due to the diverging susceptibility. Can the authors justify and comment on why the small size of their systems does not qualitatively affect their Monte Carlo results by artificially suppressing these long-wavelength fluctuations due to the small size of the numerical box?

2-Figure 1 shows the predicted critical polarisation as vertical lines. This closely matches the positions of the cusps for the HF and Popov theories, signalling the BEC transition for the minority component. However, the Monte Carlo data in this figure looks smooth. Can the authors comment whether this is also approximately the transition point for the Monte Carlo simulations, and what evidence do they have for this?

3-Some of the statements made seem a little strong to me, for example in the conclusion "We can rule out a ferromagnetic transition predicted to occur at finite temperature by perturbative approaches" or in the abstract "Results on the magnetic susceptibility show that the conditions for phase separation are not modified from the zero temperature case.". While I do agree that the results presented show clear differences between the theories, and are consistent with these statements, I am not convinced that they have definitely proven them. I say this partly because only a perfectly balanced mixture is considered, with a focus mostly on a single temperature, and some of the error bars on the Monte Carlo are quite large.

4-a little before section 3.1 begins there is the sentence "and the stable minimum at finite p predicted by Popov theory is suppressed as $g^{3/2}$ and furthermore the minimum is shifted towards higher temperatures.". Could the authors clarify what is meant, for example I do not understand which minimum is shifted to higher temperatures.

---

## Round 2 · Author Response

We thank the First Referee for their essentially positive assessment on the relevance and validity of our manuscript. In the revised version and in the reply to the Referee, we address the three comments raised by them.

---

## Round 2 · List of Changes

To address the first comment made by the First Referee, we have rewritten the paragraph at the end of the section where we discuss Fig.2 and Fig.3., as follows:

> “From these results we conclude that, in contrast to HF and Popov predictions, the magnetic susceptibility depends very little on the temperature, and the conditions for phase separation seem to remain the same as at $T=0$. In fact, if $g_{12}<g$, our results indicate that the only thermodynamically stable phase is the paramagnetic state at $p=0$. A ferromagnetic state forms when $g_{12}>g$ and the effect of temperature is to reduce the equilibrium polarization from the $p=1$ value achieved only at zero temperature. This is found at a high temperature not far from the BEC transition point and we expect the same to be true also for lower temperatures, where thermal effects not captured by the mean-field description should play a minor role. In this respect one should also notice that higher order interaction effects at T=0 do not change the critical value g_{12}=g for the onset of ferromagnetism (see Ref.[16]). As an additional remark, we point out that our results do not exclude a non trivial interplay between ferromagnetic and critical fluctuations in the close vicinity of the transition point. To carefully investigate these effects would require a much deeper analysis of the shift of the transition point in interacting mixtures beyond the scope of this work. Furthermore, we expect the simple $T=0$ scenario to hold also at densities lower than $na^3=10^{-4}$. We checked…”

To address the second comment raised by the First Referee, we have included the following comment:

> “This choice of parameters and, in particular, the choice of temperature emphasizes thermal effects in HF and Popov theories yielding important corrections to the $T=0$ magnetic susceptibility. We also note that finite-size effects in PIMC simulations of the free energy are negligible if one increases further the total number of particles.”

To address the third comment raised by the First Referee, we included the following statement with the related additional references:

> “The Hartree-Fock and Popov theories of repulsive binary Bose mixtures at finite temperature are described in details in Refs.~\cite{PhysRevLett.123.075301, PhysRevA.102.063303}. We note that Popov’s theory is also known as the finite temperature extension of Beliaev’s approach and includes the important contribution of anomalous fluctuations to thermodynamic quantities \cite{Phys. Rev. A 97, 033627 (2018); Phys. Rev. A 104, 023310 (2021)}. Here we report…”

---

## Round 3 · Referee Report · Anonymous (Referee 2) · 2023-9-7

Report

The authors have adequately addressed all of my comments and I'm now happy to recommend publication.

---

## Round 3 · Author Response

We thank the Referee for the positive evaluation of our work and for the useful comments which help us improve the manuscript. We address below the points raised by the Referee, listing the corresponding changes made to the manuscript.

  1. We agree with the referee that finite size effects are present. However, while they play a substantial role in the computation of correlation functions and susceptibilities, thermodynamic quantities such as chemical potential and free energy are less affected by them. This is especially true when looking at free energy differences. As mentioned in the text (pag. 6), we have checked that finite size effects are very small, and the overall picture does not change at larger sizes (chemical potential and free energy data have been checked against systems with double the size reported in the manuscript). For the computation of the thermodynamic properties of balanced mixtures we have instead performed the extrapolation to the thermodynamic limit considering sizes up to 512 particles. We have also looked at the particle positions snapshots for large systems (N=8000).

  2. The gray vertical lines represent the critical polarizations for the ideal gas. The effect of interactions on the critical temperature is very small (see ref. [26]), therefore the BEC transition for the minority component is well approximated by the non-interacting critical densities that give the vertical lines in fig.1. The Monte Carlo results across the transition do appear smoother than what predicted by perturbative theories, this also happens for the single component case (see fig.9 in Appendix A.2), where the Monte Carlo data reproduces the results from the universal relations of ref.[26]. However we do not have the sufficient resolution around the transition to investigate the presence of a cusp.

  3. We have changed the following sentences, softening the claims concerning the ferromagnetic transition as suggested by the Referee.

Third sentence of the abstract: "Results on the magnetic susceptibility show ..." has been changed to "Results on the magnetic susceptibility suggest ..."

Second sentence of the conclusions: "We can rule out a ferromagnetic transition predicted to occur at finite temperature by perturbative approaches and we find good agreement with the magnetic susceptibility from simple mean-field theory at zero temperature." has been changed to "For the values of the parameters considered in the simulations we do not find the ferromagnetic transition predicted to occur at finite temperature by perturbative approaches and we find good agreement with the magnetic susceptibility from simple mean-field theory at zero temperature. We further argue that a similar conclusion is expected to hold for lower values of the gas parameter."

  1. Popov theory predicts a minimum in the free energy with a depth that scales as g^{3/2} as the gas parameter na^3 goes to zero. Moreover, this minimum in the free energy is shifted at higher temperatures when the gas parameter is reduced, suggesting that even in this regime critical fluctuations would invalidate the results from HF and Popov theories. We have improved the relevant text as follows:

Last sentence before Section 3.1: "We checked numerically that for vanishing gas parameter the free energy difference between the $p=0$ state and the stable minimum at finite $p$ predicted by Popov theory is suppressed as $g^{3/2}$ and furthermore the minimum is shifted towards higher temperatures. As a consequence, we expect critical fluctuations to play a major role in the magnetic response of the mixture also in the regime of extremely low densities." has been changed to "Numerical checks show that for vanishing gas parameter the free energy difference between the $p=0$ state and the stable minimum at finite $p$ predicted by Popov theory is suppressed as $g^{3/2}$ and furthermore the minimum is shifted towards higher temperatures occurring closer to the transition point. As a consequence, we expect critical fluctuations to play a major role in the magnetic response of the mixture also in the regime of extremely low densities, thereby invalidating the predictions of Popov theory."

---

## Round 3 · List of Changes

1) Third sentence of the abstract: "Results on the magnetic susceptibility show ..." has been changed to "Results on the magnetic susceptibility suggest ..."

2) Second sentence of the conclusions: "We can rule out a ferromagnetic transition predicted to occur at finite temperature by perturbative approaches and we find good agreement with the magnetic susceptibility from simple mean-field theory at zero temperature." has been changed to "For the values of the parameters considered in the simulations we do not find the ferromagnetic transition predicted to occur at finite temperature by perturbative approaches and we find good agreement with the magnetic susceptibility from simple mean-field theory at zero temperature. We further argue that a similar conclusion is expected to hold for lower values of the gas parameter."

3) Last sentence before Section 3.1: "We checked numerically that for vanishing gas parameter the free energy difference between the $p=0$ state and the stable minimum at finite $p$ predicted by Popov theory is suppressed as $g^{3/2}$ and furthermore the minimum is shifted towards higher temperatures. As a consequence, we expect critical fluctuations to play a major role in the magnetic response of the mixture also in the regime of extremely low densities." has been changed to "Numerical checks show that for vanishing gas parameter the free energy difference between the $p=0$ state and the stable minimum at finite $p$ predicted by Popov theory is suppressed as $g^{3/2}$ and furthermore the minimum is shifted towards higher temperatures occurring closer to the transition point. As a consequence, we expect critical fluctuations to play a major role in the magnetic response of the mixture also in the regime of extremely low densities, thereby invalidating the predictions of Popov theory."

---

## Editorial Decision

published